# ATP hydrolysis by the viral RNA sensor RIG-I prevents unintentional recognition of self-RNA

Charlotte Lässig[1], Sarah Matheisl[1†], Konstantin MJ Sparrer[2†‡],
Carina C de Oliveira Mann[1†], Manuela Moldt[1], Jenish R Patel[3,4], Marion Goldeck[5],
Gunther Hartmann[5], Adolfo García-Sastre[3,4,6], Veit Hornung[7],
Karl-Klaus Conzelmann[2], Roland Beckmann[1,8], Karl-Peter Hopfner[1,8]*

[1]Gene Center, Department of Biochemistry, Ludwig Maximilian University of Munich, Munich, Germany; [2]Max von Pettenkofer-Institute, Gene Center, Ludwig Maximilian University of Munich, Munich, Germany; [3]Department of Microbiology, Icahn School of Medicine at Mount Sinai, New York, United States; [4]Global Health and Emerging Pathogens Institute, Icahn School of Medicine at Mount Sinai, New York, United States; [5]Institute for Clinical Chemistry and Clinical Pharmacology, University Hospital Bonn, University of Bonn, Bonn, Germany; [6]Department of Medicine, Division of Infectious Diseases, Icahn School of Medicine at Mount Sinai, New York, United States; [7]Institute of Molecular Medicine, University Hospital Bonn, University of Bonn, Bonn, Germany; [8]Center for Integrated Protein Science Munich, Munich, Germany

*For correspondence: hopfner@
genzentrum.lmu.de

†These authors contributed
equally to this work

Present address: ‡ Department
of Microbiology and
Immunobiology, Harvard Medical
School, Boston, United States

Competing interest: See
page 16

Reviewing editor: Stephen C
Kowalczykowski, University of
California, Davis, United States

**Abstract** The cytosolic antiviral innate immune sensor RIG-I distinguishes 5′ tri- or diphosphate containing viral double-stranded (ds) RNA from self-RNA by an incompletely understood mechanism that involves ATP hydrolysis by RIG-I's RNA translocase domain. Recently discovered mutations in ATPase motifs can lead to the multi-system disorder Singleton-Merten Syndrome (SMS) and increased interferon levels, suggesting misregulated signaling by RIG-I. Here we report that SMS mutations phenocopy a mutation that allows ATP binding but prevents hydrolysis. ATPase deficient RIG-I constitutively signals through endogenous RNA and co-purifies with self-RNA even from virus infected cells. Biochemical studies and cryo-electron microscopy identify a 60S ribosomal expansion segment as a dominant self-RNA that is stably bound by ATPase deficient RIG-I. ATP hydrolysis displaces wild-type RIG-I from this self-RNA but not from 5′ triphosphate dsRNA. Our results indicate that ATP-hydrolysis prevents recognition of self-RNA and suggest that SMS mutations lead to unintentional signaling through prolonged RNA binding.

## Introduction

The innate immune system provides a rapid initial reaction to invading pathogens and also stimulates the adaptive immune system (*Iwasaki and Medzhitov, 2015*). Pattern recognition receptors (PRRs) of the innate immune system sense pathogen- or danger-associated molecular patterns (PAMPs or DAMPs) and trigger molecular cascades that together initiate and orchestrate the cellular response through activation of e.g. interferon regulatory factors and nuclear factor κB (*Brubaker et al., 2015*; *Pandey et al., 2015*; *Wu and Chen, 2014*).

Retinoic-acid inducible gene I (RIG-I), melanoma differentiation-associated gene 5 (MDA5) and laboratory of physiology and genetics 2 (LGP2) are three structurally related PRRs – denoted RIG-I

**eLife digest** Living cells produce long, strand-like molecules of RNA that carry the instructions needed to make proteins. Viruses also make use of RNA molecules to hijack an infected cell's protein-production machinery and create new copies of the virus. RNA molecules from viruses have a number of features that distinguish them from a cell's own RNAs, and human cells contain receptors called RLRs that can start an immune response whenever they detect viral RNAs. All of these receptors break down molecules of ATP, a process that releases useable energy. However, so far it is not understood how this activity helps the receptors to distinguish viral RNA from the cell's own RNA molecules (called self-RNA).

Recently, some autoimmune diseases (including Singleton-Merten Syndrome) were linked to mutations in the parts of RLRs that allow the receptors to break down ATP. Now, Lässig et al. have studied the effects of specific mutations in an RLR called RIG-I in human cells. The experiments showed that mutations that disrupt RIG-I's ability to bind to ATP also prevented the receptor from becoming activated. However, mutations linked to Singleton-Merten Syndrome don't stop ATP from binding but instead slow its breakdown; this effectively locks the receptor in an ATP-bound state. Lässig et al. found that similar mutations in RIG-I caused human cells to trigger a constant immune response against the self-RNAs.

Further experiments then suggested that the breakdown of ATP helps to remove RIG-I that has bound to double-stranded sections of self-RNAs. This activity frees the receptor, making it more able to detect double-stranded viral RNAs and preventing unintentional signaling. Lässig et al. also identified a specific double-stranded section of a human RNA that may be recognized by the mutated version of RIG-I in people with Singleton-Merten Syndrome.

The next steps following on from this work are to extend the analysis to also include other RLRs and further explore the underlying mutations within the three-dimensional structures of the receptors and RNA molecules involved.

like receptors (RLRs) – that recognize cytosolic foreign RNA. RIG-I senses RNA from a broad range of viruses including measles virus and Sendai virus (both paramyxoviridae), Influenza A virus, Japanese encephalitis virus and Hepatitis C virus, whereas MDA5 is activated for example by picornavirus RNA. LGP2 has augmenting and regulatory roles in MDA5 and RIG-I dependent signaling (*Bruns et al., 2014*; *Satoh et al., 2010*; *Sparrer and Gack, 2015*).

RIG-I preferentially detects base-paired double-stranded RNA (dsRNA) ends containing either 5′ triphosphate (ppp) or 5′ diphosphate (pp) moieties (*Goubau et al., 2014*; *Hornung et al., 2006*; *Pichlmair et al., 2006*; *Schlee et al., 2009*; *Schmidt et al., 2009*) and not 2′ OH methylated at the first 5′ terminal nucleotide (*Schuberth-Wagner et al., 2015*). ppp-dsRNA arises, for example, at panhandle structures of influenza virus nucleocapsids, or during measles or Sendai virus transcription (*Liu et al., 2015*; *Weber et al., 2013*). 5′ diphosphates are found on genomic RNA of reoviruses (*Banerjee and Shatkin, 1971*). RIG-I can also detect poly-U/UC-rich dsRNA (*Schnell et al., 2012*). Ligands of MDA5 are less well characterized but include dsRNA longer than 1000 bp (*Kato et al., 2008*), higher-order dsRNA structures (*Pichlmair et al., 2009*), or AU-rich RNA (*Runge et al., 2014*).

RLRs are members of the superfamily II (SF2) of ATPases, helicases or nucleic acid translocases. RIG-I and MDA5 consist of two N-terminal tandem caspase activation and recruitment domains (2CARD), a central ATPase/translocase domain and a C-terminal regulatory domain (RD). LGP2 lacks the 2CARD module but otherwise has a similar domain architecture. Binding of RNA induces a conformational change in RIG-I. If activated, the RD binds the ppp- or pp-dsRNA end, while the SF2 domain interacts with the adjacent RNA duplex and forms an active ATPase site (*Civril et al., 2011*). In this conformation, the 2CARD module is sterically displaced from its auto-inhibited state (*Jiang et al., 2011*; *Kowalinski et al., 2011*; *Luo et al., 2011*) and can be K63-linked poly-ubiquitinated (*Gack et al., 2007*). Multiple Ub-2CARD complexes assemble to form a nucleation site for the polymerization of mitochondrial antiviral-signaling adaptor protein (MAVS) into long helical filaments (*Hou et al., 2011*; *Wu et al., 2014*; *Xu et al., 2014*). Instead of recognizing terminal structures like

RIG-I, MDA5 cooperatively polymerizes along dsRNA (*Berke and Modis, 2012*), which is suggested to trigger MAVS polymerization.

The SF2 ATPase domain plays a critical part in RIG-I activation, although the role of the ATPase activity is still debated. Mutation of the seven SF2 "helicase" motifs resulted in RLRs that are either inactive or signal constitutively (*Bamming and Horvath, 2009*; *Louber et al., 2015*). On the other hand, overexpression of the 2CARD module alone is sufficient for signaling (*Yoneyama et al., 2004*). Further studies revealed that the SF2 domain is an ATP-dependent dsRNA translocase (*Myong et al., 2009*) that can help enhance signaling by loading multiple RIG-I on dsRNA (*Patel et al., 2013*) and may execute anti-viral "effector" functions through displacement of viral proteins (*Yao et al., 2015*). Finally, RIG-I ATPase activity promotes recycling of RIG-I:dsRNA complexes in vitro, suggesting a kinetic discrimination between self and non-self RNA (*Anchisi et al., 2015*; *Louber et al., 2015*).

Several autoimmune diseases, including the Aicardi-Goutières and Singleton-Merten syndromes (SMS), were linked to single amino acid mutations in the SF2 domains of MDA5 and RIG-I (*Funabiki et al., 2014*; *Jang et al., 2015*; *Rice et al., 2014*; *Rutsch et al., 2015*). Two point mutations within the Walker A (motif I) or Walker B (motif II) of RIG-I are linked to atypical SMS and functional studies indicated constitutive RIG-I activation (*Jang et al., 2015*). Thus, these mutations have been described as a gain of function, which is puzzling considering previous mutations in motif I led to loss of RIG-I function, while mutations in motif II led to either gain or loss of function, depending on the type of mutation (*Bamming and Horvath, 2009*; *Louber et al., 2015*).

In order to clarify the role of RIG-I's ATPase in antiviral signaling and RLR associated human diseases, we engineered structure-derived and patient-identified mutations into RIG-I and tested the resulting proteins in different types of cell-based and in vitro analyses. Collectively, we find that SMS mutations phenocopy the structure-derived E373Q mutation in motif II, which is designed to trap RIG-I in an ATP-bound state. Freezing this state results in a dramatic autoimmune response because the enzyme binds self-RNA and signals. An unexpected, strongly enriched self-RNA is the ribosomal large subunit, which contains large, dsRNA expansion segments. Collectively, our results suggest that a biomedical and functional critical role of RIG-I's ATPase is to prevent spontaneous and unintended activation by self-RNA. Thus, the SF2 translocase likely increases the sensitivity of the system by reducing background signaling. Furthermore, our studies suggest that in SMS, RIG-I is trapped in an ATP-bound state and signals through self-ligands.

## Results

### Prevention of ATP hydrolysis in RIG-I leads to a constitutive activation of the interferon-β promoter by recognition of self-RNA

To address the roles of ATP binding and hydrolysis by the SF2 domain of RIG-I, we studied RIG-I variants containing structure-based mutations designed to i) prevent ATP binding and formation of a functional ATP-bound complex, ii) allow ATP binding and ATP-induced conformational changes but prevent ATP hydrolysis, or iii) disable interaction of the RNA with either the 1A or 2A domain of SF2 (*Figure 1A, B*). The structure of RIG-I in complex with RNA and ADP·BeF$_x$ served as guide for these mutations ([*Jiang et al., 2011*], PDB code 3TMI, *Figure 1B*).

In order to dissect the influence of these mutations on the ability of RIG-I to elicit downstream signaling, we used an interferon-β (IFNβ) promoter activity assay carried out in HEK 293T RIG-I KO cells (*Figure 1—figure supplement 1A,B*). Overexpressed wild-type RIG-I (wtRIG-I) is able to induce a slight activation of the IFNβ promoter, which can be further amplified by stimulation with Sendai virus defective interfering particles (SeV DIs) (*Figure 1C*). The 2CARD module (RIG-I 1-229) induced a strong activation in both non-infected and SeV DI-stimulated cells and is crucial since constructs lacking these domains (Δ2CARD, RIG-I 230-925) cannot conduct any downstream signaling. RIG-I K270I, carrying a mutation in the motif I lysine that reduces ATP binding (*Rozen et al., 1989*), signaled in neither uninfected nor SeV DIs stimulated cells, consistent with previous studies. Remarkably, the E373Q substitution in motif II had a strikingly different effect. RIG E373Q, which has a stabilized ATP-bound state by slowed-down ATP hydrolysis, strongly signaled in both non-infected and SeV DIs stimulated cells. Western blots validated correct expression of all mutants (*Figure 1—figure supplement 1C*).

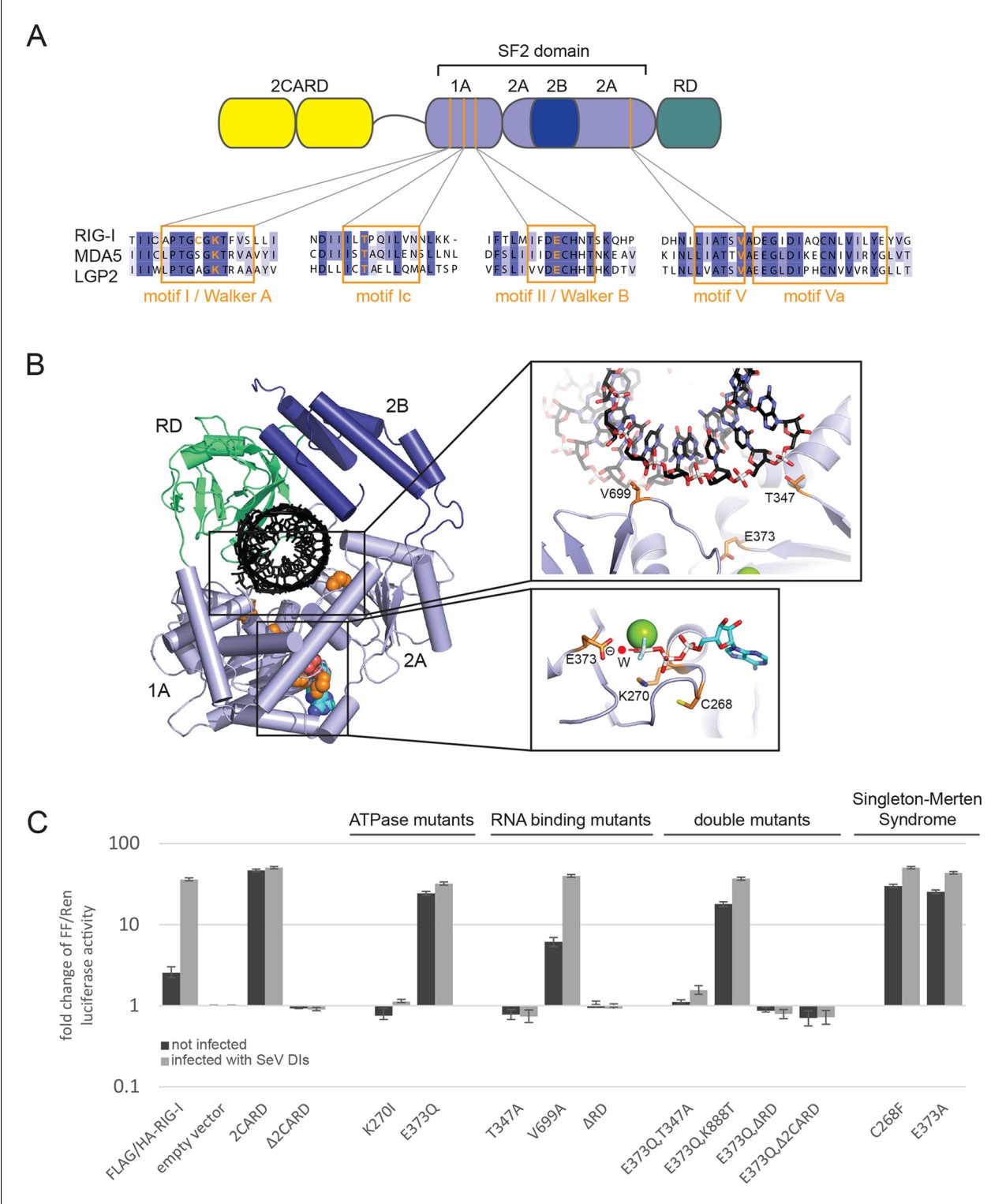

**Figure 1.** Cellular studies of RIG-I ATPase mutants in infected or non-infected cells. (**A**) Location of amino acid substitutions of RIG-I SF2 domain variants used in this study (orange lines) within different RLR helicase motifs (orange squares). (**B**) Single amino acid substitutions (orange) within the RIG-I 3D structure (PDB: 3TMI). (**C**) Fold change of interferon-β (IFNβ) promoter driven luciferase activity in uninfected HEK 293T RIG-I KO cells or in cells challenged with Sendai virus defective interfering particles (SeV DIs). Cells were co-transfected with RIG-I expression vectors and p125-luc/ pCMV-RL reporter plasmids, and infected with SeV DIs 6 hr post transfection. Firefly (FF) luciferase activities were determined in respect to Renilla (Ren) luciferase activities 24 hpi. All ratios were normalized to the empty vector control. n=3–12, error bars represent mean values ± standard deviation.

*Figure 1 continued on next page*

To rule out a "constitutive" active conformation of RIG-I E373Q due to an exposed 2CARD module (e.g. from an unfolded SF2) we performed small angle X-ray scattering with purified wtRIG-I and RIG-I E373Q demonstrating that both proteins have the same solution structure (*Figure 1—figure supplement 2A, B*). In addition, thermal unfolding assays show that the E373Q mutation does not destabilize RIG-I (*Figure 1—figure supplement 2C*). Finally, RIG-I Δ2CARD,E373Q has a dominant negative effect on signaling by RIG-I E373Q (*Figure 1*, *Figure 1—figure supplement 2D, E*). Taken together, these data show that RIG-I E373Q is neither destabilized nor constitutively active, suggesting it needs productive RNA interactions.

To test whether E373Q signals in non-infected (and perhaps also infected cells) because of interaction with self-RNA, we additionally introduced mutations in various RNA binding sites, in particular a ΔRD variant (RIG-I 1-798) and mutations in two RNA-interacting residues in domains 1A (T347A) and 2A (V699A) of SF2. The single mutation RIG-I T347A did not signal in either infected or non-infected cells, showing that the interaction of RNA with this specific amino acid is critical for signaling (*Figure 1C*). Interestingly, we find that the single mutation V699A slightly increases the signaling activity of RIG-I in non-infected cells (*Figure 1C*), which could be explained by a putative reduction of translocation activity instead of a prevention of RNA binding to SF2 (see discussion). Finally, deletion of the regulatory domain (ΔRD) inactivates signaling in both infected and non-infected cells as previously observed (*Cui et al., 2008*). As expected, both combination mutants RIG-I E373Q,T347A and RIG-I E373Q,ΔRD failed to signal in both SeV DIs infected and non-infected cells. These data show that the increased immunostimulatory effect of E373Q requires a productive RNA interaction of SF2 and RD.

Since RD is also required for the displacement of the 2CARD module from SF2, we additionally analyzed a point mutation in RD. K888 mediates triphosphate binding in RD and mutations in this residue inactivate recognition of viral RNA (*Cui et al., 2008*; *Wang et al., 2010*). Of note RIG-I E373Q,K888T is still constitutively active in non-infected cells. This effect indicates that the increased signaling capacity on endogenous RNA is independent from the ppp-dsRNA or pp-dsRNA epitopes that RIG-I recognizes on viral RNA via the RD.

Finally, we addressed the effect of the Singleton-Merten mutations C268F and E373A. E373A is at the same position as our structure-derived E373Q mutant. Consistent with this, we observed that this substitution leads to a constitutive activation of the IFNβ promoter (*Jang et al., 2015*) (*Figure 1C*). Interestingly, although C268 is located in motif I, it also leads to constitutive signaling, whereas motif I mutation of K270 (which coordinates the β-phosphate of ATP) blocks ATP binding and renders RIG-I inactive. Thus, mutation of the non-ATP binding C268 in motif I appears to phenocopy a mutation that prevents ATP hydrolysis.

In summary, our studies show that signaling of RIG-I requires both ATP and RNA binding. ATP hydrolysis, on the other hand, appears to be critical to dissolve the signaling state and to prevent activation of RIG-I by self-RNA.

## RIG-I ATP hydrolysis defective mutant E373Q shows increased interaction with ribosomal RNA

We hypothesized that E373Q traps RIG-I in an ATP bound high affinity conformation that is activated already by self-RNA. To test this idea, we immunoprecipitated RIG-I and its mutants from non-infected HEK 293T RIG-I KO cells or cells infected with measles or Sendai virus and analyzed the co-purified RNA molecules. Regardless of whether co-purified from infected or non-infected cells, the amount of RNA recovered from RIG-I E373Q was about 3 times higher than that from RIG-I (*Figure 2A*). Similarly increased amounts of RNA co-purified with the SMS mutants C268F and E373A from uninfected cells, reflecting the same altered RNA binding properties as in RIG-I E373Q (*Figure 2—figure supplement 1A*).

When analyzed on a Bioanalyzer RNA chip or on agarose gels, we found that the increased amount of RNA is to a large extent due to the presence of 28S rRNA, while 18S rRNA remains unaltered (*Figure 2B*). Control analysis of the total RNA content ruled out an alteration of ribosome subunit ratio in RIG-I E373Q transfected cells (*Figure 2C*). Both increased amount of RNA and specific enrichment of 28S rRNA were also observed for the equivalent MDA5 E444Q Walker B mutant (*Figure 2—figure supplement 1B, C*).

In order to determine the immunostimulatory potential of the RNA co-purified from virus-infected cells, we back-transfected the RNA into HEK 293T ISRE-FF/RFP reporter cells (which contain

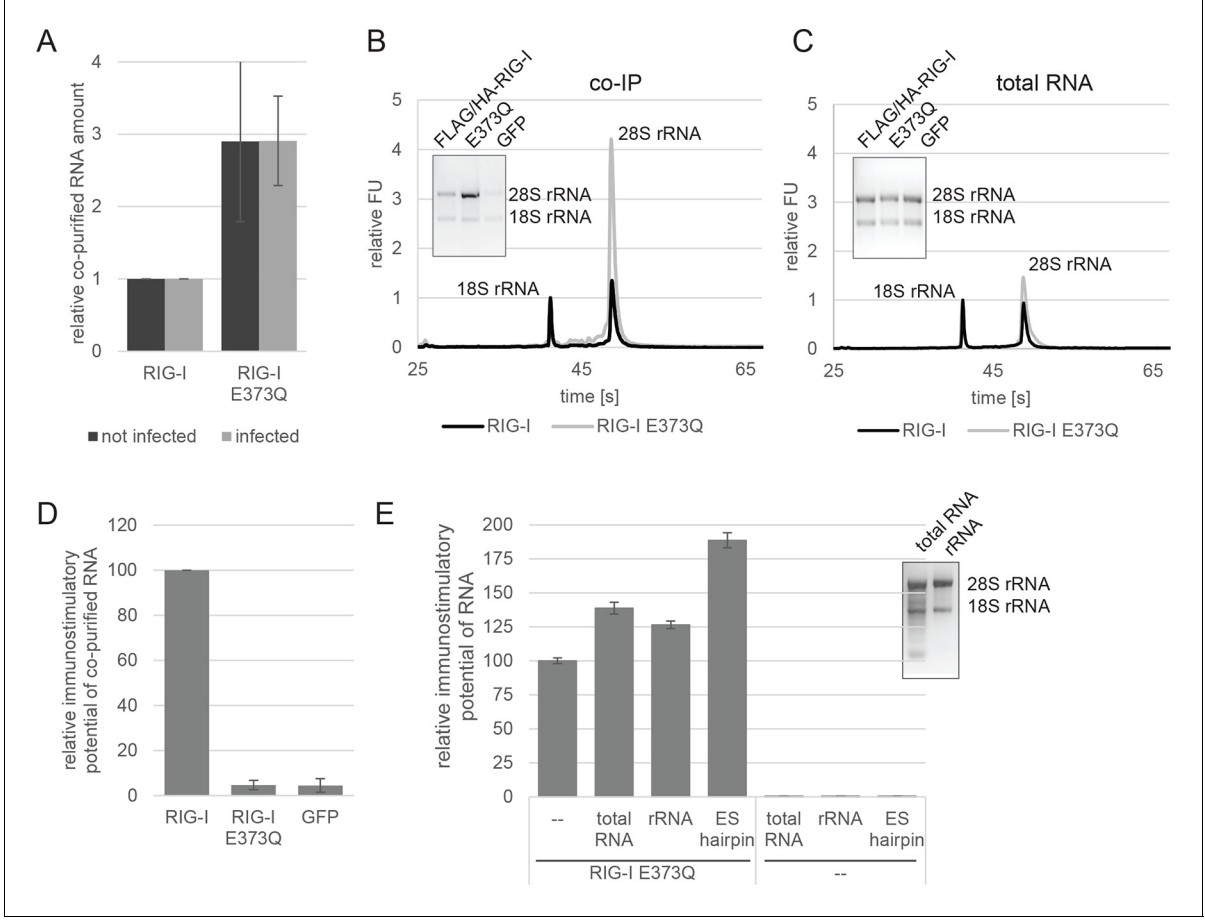

**Figure 2.** RIG-I ATP hydrolysis defective mutant E373Q recognizes the 60S ribosomal subunit in vivo. (A) Relative RNA amount co-purified with overexpressed RIG-I or RIG-I E373Q from virus infected or non-infected HEK 293T RIG-I KO cells. n=4 (infected) or n=10 (non-infected), error bars represent mean values ± standard deviation. (B) Bioanalyzer evaluation and agarose gel separation of RNA co-purified with overexpressed RIG-I or RIG-I E373Q from non-infected HEK 293T RIG-I KO cells. Curves are normalized in respect to 18S rRNA peaks. (C) Bioanalyzer evaluation and agarose gel separation of total RNA content of non-infected HEK 293T RIG-I KO cells overexpressing RIG-I or RIG-I E373Q. Curves were normalized as in panel B. (D) Immunostimulatory potential of co-purified RNA from RIG-I, RIG-I E373Q or GFP overexpressed in measles virus (MeV), MeV-Cko-ATU-Cs or Sendai virus Cantell (SeV) infected HEK 293T RIG-I KO cells. RNA was back-transfected into HEK 293T ISRE-FF/RFP cells together with pTK-RL transfection control. Firefly luciferase (FF) activities were determined 24 hr after transfection in respect to Renilla luciferase (Ren) activity and were normalized to the immunostimulatory potential of RIG-I associated RNA. n=4, error bars represent mean values ± standard deviation. (E) Immunostimulatory potential of endogenous RNA in cells overexpressing RIG-I E373Q. RNA was co-transfected into HEK 293T RIG-I KO cells together with a RIG-I E373Q expression vector and p125-luc/ pCMV-RL reporter plasmids. FF luciferase activities were determined in respect to Ren luciferase activities 24 hr after transfection. All ratios are normalized to the RIG-I E373Q control without RNA stimulation. Purified RNA was in addition analyzed on agarose gels. n=3, error bars represent mean values ± standard deviation.

The following figure supplements are available for figure 2:

**Figure supplement 1.** Analysis of RNA co-purified with RIG-I SMS or MDA5 variants.

**Figure supplement 2.** Assay for defining the immunostimulatory potential of different RNAs.

**Figure supplement 3.** Immunostimulatory potential of co-purified RNA from Sendai virus Cantell (SeV) infected cells.

endogenous RIG-I, see *Figure 2—figure supplement 2A*). RNA co-purified with wtRIG-I and RIG-I lacking the 2CARD module induced an immune response in these cells (*Figure 2—figure supplement 3A*). RNA co-purified with RIG-I K270I (ATP binding deficient) and V699A (putative translocation deficient) was also able to stimulate the ISRE reporter in an amount comparable to wtRIG-I, indicating no altered RNA binding properties in these mutants under virus infected conditions. In

contrast, RNA that co-purified with the RNA-binding deficient RIG-I T347A (mutation in SF2 domain), RIG-I K858E (mutation in RD domain that reduces triphosphate recognition) or RIG-I ΔRD poorly stimulated the ISRE promoter and probably represents background RNA (*Figure 2—figure supplement 3A*). These data suggest that RIG-I recognizes immunostimulatory RNA via the SF2 and RD domains, but does not require ATP binding for this process. ATP binding is necessary, however, because RIG-I K270I expression alone does not stimulate the IFNβ promoter (compare with *Figure 1C*). Interestingly, RNA co-purified with RIG-I E373Q failed to induce reporter gene expression (*Figure 2D*, *Figure 2—figure supplement 3A*). Thus, despite the observation that RIG-I E373Q co-purifies with approximately threefold more RNA than wtRIG-I from infected cells, the co-purified RNA is not immunostimulatory in a wtRIG-I background. However, cells that transiently express RIG-I E373Q can be further stimulated by transfection of total RNA extracts and purified ribosomal RNA (*Figure 2E*), suggesting that ribosomal RNA can activate RIG-I E373Q. Cells lacking wtRIG-I or RIG-I E373Q on the other hand do not respond to those RNAs. We conclude that host-RNA, which does not activate wtRIG-I, can apparently compete with viral RNA for RIG-I E373Q.

In order to verify a higher affinity of the RIG-I ATP hydrolysis defective mutant towards ribosomal RNA, we purified full-length human RIG-I and RIG-I E373Q, as well as human 80S ribosomes, and tested for a direct interaction. We confirmed that while both RIG-I E373Q and the wild-type protein are able to bind ATP, only wtRIG-I can hydrolyze ATP (*Figure 3A, B*). We subsequently conducted sedimentation assays via ultra-centrifugation of sucrose cushions loaded with 80S ribosomes that have been pre-incubated with wtRIG-I or RIG-I E373Q in presence or absence of ATP or the non-hydrolysable ATP analogue ADP·BeF$_3$. In presence of ATP a minor binding of wtRIG-I to the ribosome could be observed, whereas RIG-I E373Q bound in a near stoichiometric manner. In absence of ATP or in presence of ADP·BeF$_3$ binding of wtRIG-I was greatly enhanced and showed similar levels compared to RIG-I E373Q (*Figure 3C*).

We next analyzed RIG-I E373Q:80S ribosome complexes by cryo-electron microscopy and single particle 3D reconstruction (*Figure 3D*). The average resolution was estimated to be 17.7 Å based on the Fourier shell correlation cut-off criterion at 0.5. When compared with the reconstruction of the human 80S ribosome alone (*Figure 3E*), the ribosome:RIG-I E373Q complex revealed an additional density located at rRNA expansion segment (ES) 7L, which is located at the back of the large ribosomal subunit. Calculation of a statistical difference map between the two reconstructions confirmed that this distinct region contained significant additional density (*Figure 3F*). Human ribosomes contain several long, G:C rich, base-paired RNA expansion segments forming large tentacle-like hairpin structures of substantial double-stranded nature (*Anger et al., 2013*). A large part of the double-stranded RNA in these segments is not covered by ribosomal proteins and accessible for cytosolic proteins. The crystal structure of ADP·BeF$_x$-bound RIG-I △2CARD:RNA complex ((*Jiang et al., 2011*), PDB code 3TMI) fits well into the density observed at ES7L and is located at the root of the solvent exposed portion of helix A of ES7L that contains a contiguous stretch of seven G:C/C:G base pairs (*Figure 3G*).

In summary, we conclude that stabilizing the ATP-bound state of RIG-I induces a conformation where RIG-I binds to ribosomes, presumably at exposed dsRNA expansion segments.

## Specificity of RIG-I towards double-stranded RNA is increased in presence of ATP

To further evaluate the role of ATP binding and hydrolysis of RIG-I we performed electrophoretic mobility shift assays (EMSAs), fluorescence anisotropy experiments and ATP hydrolysis assays in presence and absence of ATP or ADP·BeF$_3$ with different RNAs. These RNAs mimic different types of endogenous or viral RNAs and help dissecting contributions of RD's binding to the RNA end and SF2's binding to the stem. In addition to a 24mer or 12mer blunt-ended dsRNA or ppp-dsRNA (*Goldeck et al., 2014*), we also used a 60 nucleotide hairpin RNA (denoted as ES hairpin) derived from the ribosomal expansion segment ES7L, which contains several bulges and a non-pairing end (*Figure 4—figure supplement 1A*). The hairpin at one end and the added Y-structure at the other end are used to minimize RNA end binding by RIG-I's RD because RD has a high affinity for blunt RNA ends.

RIG-I and RIG-I E373Q bound to the 24mer blunt ended dsRNA with a slightly higher affinity in presence of ATP or ADP·BeF$_3$ than in its absence (*Figure 4A*), suggesting that ATP binding to the SF2 domain positively contributes to the overall affinity in addition to RD. A similar result was

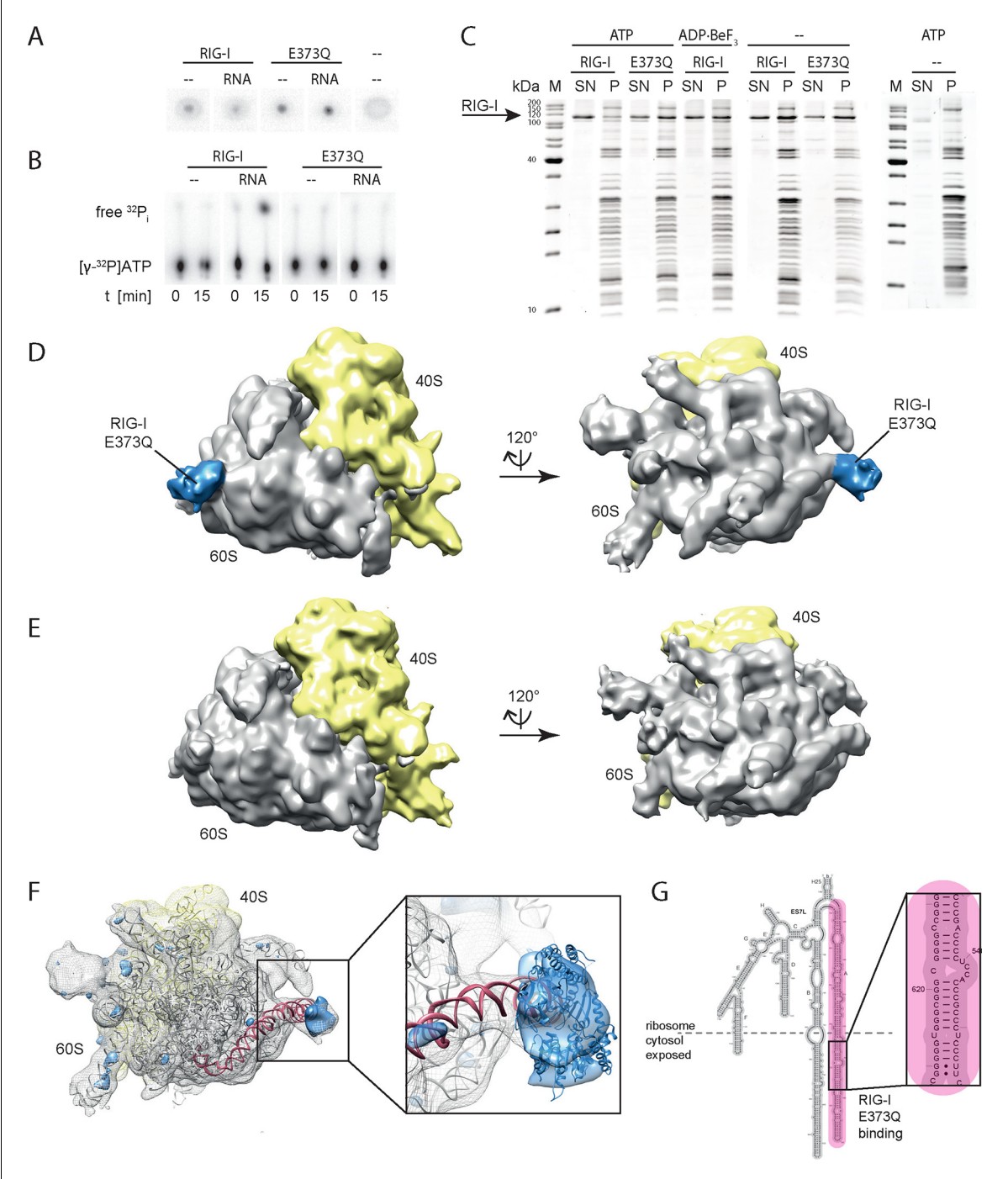

**Figure 3.** RIG-I ATP hydrolysis defective mutant E373Q recognizes the 60S ribosomal subunit in vitro. (**A**) DRaCALA ATP binding assay of RIG-I or RIG-I E373Q in presence or absence of RNA. (**B**) ATP hydrolysis assay of RIG-I or RIG-I E373Q in presence and absence of RNA. (**C**) Binding studies of human 80S ribosomes with RIG-I or RIG-I E373Q in presence or absence of ATP or ADP·BeF$_3$. Pre-formed complexes were separated on sucrose cushions via ultracentrifugation and pellet (P) as well as supernatant (SN) fractions were analyzed by SDS-PAGE. (**D**) Side views of a cryo-EM reconstruction of RIG-I E373Q (blue) bound to the human 80S ribosome (yellow: 40S subunit, gray: 60S subunit). Data was low pass-filtered at 15 Å. (**E**) Side views of a cryo-EM reconstruction of the human 80S ribosome without prior RIG-I E373Q incubation. Data filtering and color coding as in panel D. (**F**) Statistical difference map (left, σ = 2) of cryo-EM reconstructions in panels D and E reveals a significant additional density at expansion segment 7L A (ES7L-A, pink) into which RIG-I (PDB 3TMI) can be fitted (right, σ = 1.51). (**G**) Secondary structure map of the 28S rRNA ES7L (derived from (**Anger et al., 2013**) and zoom into RIG-I E373Q binding area. ES7L-A is indicated in pink (as in panel **F**).

obtained when we used a 12mer dsRNA in fluorescence anisotropy experiments in order to further dissect the influence of different RNA ends (*Figure 4B*). Interestingly, the positive effect of ATP was not observed when we used the corresponding ppp-dsRNA 12mer (*Figure 4C*), most likely because the RD dominates RNA binding under these conditions. Thus, it is plausible that RIG-I dissociates from unphosphorylated RNA termini with an increased rate after ATP hydrolysis than from triphosphorylated termini.

We next tested the role of ATP on binding of wtRIG-I, RIG-I E373Q, RIG-I T347A,E373Q and the SMS variant RIG-I C268F to the ES hairpin RNA mimicking the base of the ribosomal ES7L. In presence of ATP we observed moderately increased binding of RIG-I E373Q and of RIG-I C268F to this hairpin, however wtRIG-I displayed a strikingly opposing effect (*Figure 4D*, *Figure 4—figure supplement 1C*). For this RNA, ATP reduced rather than increased the affinity of wtRIG-I. The addition of ADP·BeF$_3$ to RIG-I could reconstitute the high affinity state of RIG-I E373Q. The RIG-I T347A,E373Q double mutant, on the other hand, showed binding affinities similar to RIG-I in presence of ATP, probably caused by residual binding of RD (*Figure 4D*).

Consistent with this, the ES hairpin RNA could induce signaling in RIG-I E373Q transfected HEK 293T RIG-I KO cells (*Figure 2F*) and could also stimulate the ATPase activity of RIG-I △2CARD, and to a lesser extent wtRIG-I (which is auto-inhibited by the 2CARD module) (*Figure 5A*, *Figure 5—figure supplement 1A*). A comparable stimulatory effect on the ATPase activity of RIG-I could also be detected with whole human ribosomes (*Figure 5A*). Control assays with the ATP hydrolysis defective mutants RIG-I E373Q and RIG-I T347A,E373Q confirmed the lacking ability of those proteins to hydrolyze ATP even in the presence of triphosphorylated RNA (*Figure 5A*, *Figure 5—figure supplement 1B*).

In summary, our results show that ATP hydrolysis leads to a moderately increased binding of RNA containing base-paired ends, but decreased binding of RNA lacking base-paired ends. These in vitro data are also consistent with our co-immunopurification studies of RNA from cells, where we observed that the ATP hydrolysis deficient RIG-I E373Q mutant co-purified with increased amounts of endogenous RNA.

## Discussion

Here we show that mutations that slow down or inhibit RIG-I's ATPase lead to an increased interaction of RIG-I with endogenous RNA, including double-stranded RNA expansion segments of the human large ribosomal subunit. Our results suggest that RIG-I's ATPase confers specificity to viral RNA by preventing signaling through the abundant background of self-RNA and provide a molecular framework for understanding the pathology of atypical Singleton-Merton syndrome.

Recently, several autoimmune diseases, including the Aicardi-Goutières and Singleton-Merten syndromes, have been linked to RLRs through whole exome sequencing, which discovered single amino acid mutations that are mostly found within the ATPase domain of RLRs (*Jang et al., 2015*; *Rice et al., 2014*; *Rutsch et al., 2015*). Increased interferon levels suggest that an increased activation of MDA5 or RIG-I underlies the molecular pathology of these diseases. Indeed we find that not only E373Q, consistent with recent results, leads to an increased activation of RIG-I in non-infected cells, but also the SMS mutations E373A and C268F (*Jang et al., 2015*) (*Figure 1C*). While this could have been expected for E373A, because of its similarity to E373Q, the increased immunostimulatory effect of C268F in motif I comes as a surprise. Prior mutations in motif I studied by others and us led to an inactivation of RIG-I, rather than constitutive activation. The precise structural reason for the increased signaling of C268F needs to be addressed in future studies, but our co-immunoprecipitation and in vitro binding assay results suggest that this mutation may also lock RIG-I in an RNA-bound, active conformation (*Figure 2—figure supplement 1A*, *Figure 4—figure supplement 1C*).

Mutational and biochemical analyses previously suggested a kinetic model for RIG-I's specificity towards viral RNA, where the ATP-dependent recycling helps to discriminate ppp-dsRNA from endogenous RNA (*Anchisi et al., 2015*; *Louber et al., 2015*; *Runge et al., 2014*) (*Figure 6A*). Our studies show that, in case of base-paired triphosphate containing RNA ends, the RIG-I RD dominates binding. Although RIG-I's ATPase is very active, we do not see a strong effect of ATP on the affinity for the RNA (*Figure 4C*, *Figure 5A*). ATP hydrolysis may under the assayed conditions not efficiently displace RIG-I from ppp-dsRNA because RD might prevent full dissociation even after ATP-hydrolysis displaced SF2. Importantly, ATP reduces the affinity towards self-RNA containing a duplex region

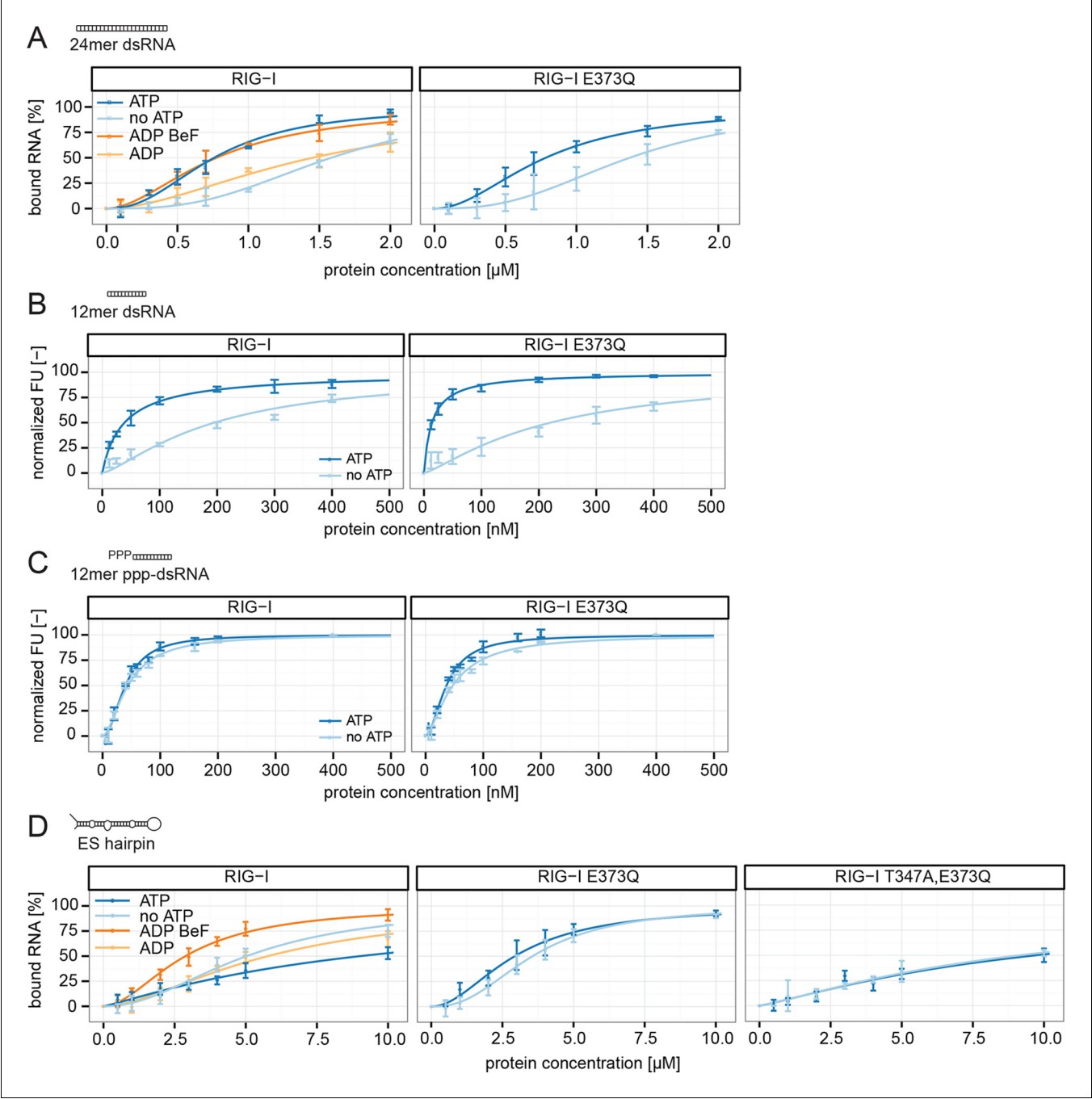

**Figure 4.** RIG-I's ATP hydrolysis enhances RNA end recognition and removes RIG-I from RNA stems. (**A**) Quantification of electrophoretic mobility shift assays of RIG-I or RIG-I E373Q incubated with 24mer dsRNA in presence or absence of ATP, ADP or ADP·BeF₃ (compare with *Figure 4—Figure supplement 1B*). (**B**) Fluorescence anisotropy changes measured by titrating RIG-I or RIG-I E373Q in presence or absence of ATP into solutions containing fluorescently labeled 12mer dsRNA. (**C**) Fluorescence anisotropy changes measured by titrating RIG-I or RIG-I E373Q in presence or absence of ATP into solutions containing fluorescently labeled 12mer ppp-dsRNA. (**D**) Quantification of electrophoretic mobility shift assays of RIG-I, RIG-I E373Q or RIG-I T347A, E373Q incubated with an RNA hairpin derived from helix A of the human ribosome expansion segment 7L (ES hairpin) in presence or absence of ATP, ADP or ADP·BeF₃ (compare with *Figure 4—Figure supplement 1C*). All binding curves were fitted using the LL.2 function of the R drc package (*Cedergreen et al., 2005*). n=3-6, error bars represent mean values ± standard deviation.

The following figure supplement is available for figure 4:

**Figure supplement 1.** Design of the ribosomal expansion segment derived hairpin RNA, EMSA raw figures and control experiments with RIG-I C268F SMS mutant.

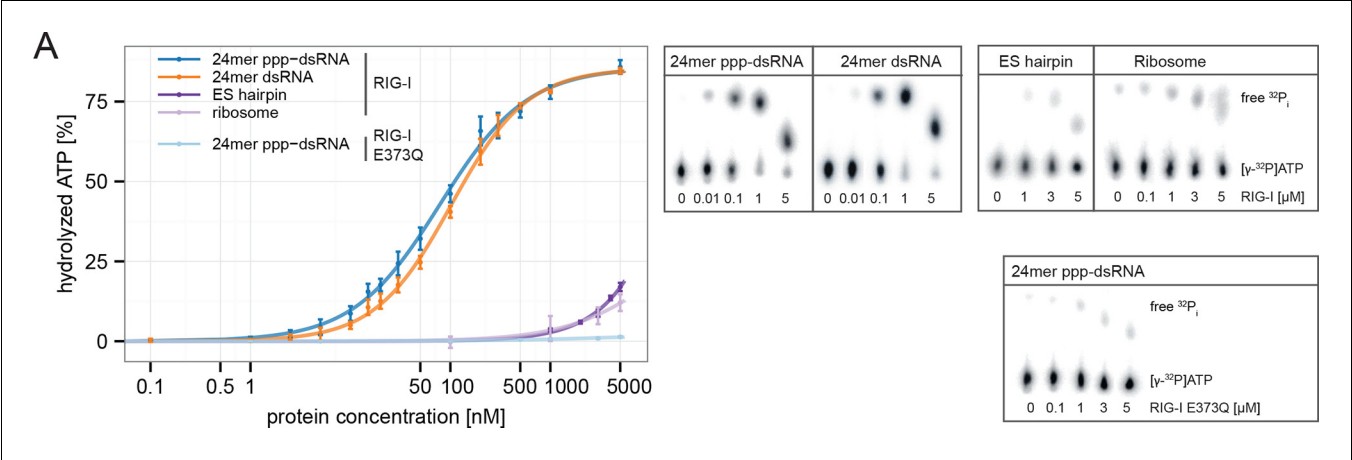

**Figure 5.** RIG-I's ATPase activity correlates with its RNA binding affinity. (**A**) Quantification of hydrolyzed [γ-$^{32}$P]ATP by RIG-I or RIG-I E373Q in presence of different RNA substrates. Reactions were allowed to proceed for 20 min at 37 °C and free phosphate was separated from ATP via thin layer chromatography. Spots corresponding to labeled ATP and labeled P$_i$ were quantified using ImageJ. All curves were fitted using the LL.2 function of the R drc package. n=3, error bars represent mean values ± standard deviation.

The following figure supplement is available for figure 5:

**Figure supplement 1.** RIG-I's 2CARD module reduces the ATP hydrolysis activity.

but not a "proper" ppp-dsRNA end (*Figure 4D*). Thus, if RD is unable to tether RIG-I to ppp-dsRNA ends the ATPase could rapidly remove RIG-I from RNA duplex regions via its translocase and therefore prevents an autoimmune response towards self-RNA (*Figure 6B*). Our cellular studies are consistent with this biochemical observation, because a point mutation in K888, a residue that is critical for recognizing ppp-dsRNA ends, did not reduce the constitutive activation of ATP hydrolysis-deficient RIG-I (*Figure 1C*). However, RD and ATP binding are clearly important for signaling, as shown by △RD and K270I mutations by us and others (*Louber et al., 2015*) (*Figure 1C*), suggesting that a ring-like, ATP-bound structure is also involved in signaling caused by self-RNA (*Figure 6C*). In this conformation, the RD likely helps to displace the 2CARD module from the SF2 domain but may not have a high affinity for the RNA itself. Of note, the mutation in V699 of motif V also leads to increased constitutive signaling (*Figure 1C*). A plausible explanation could be that this mutation in RecA2 decouples RNA-binding induced ATP hydrolysis from translocation or displacement of RNA. In summary, our results suggest a model where RIG-I's translocase removes SF2 from dsRNA, perhaps at nearby bulges, unless high-affinity binding by the RD on RNA ends containing di- or triphosphates tethers RIG-I despite ATP-hydrolysis and leads to repeated or prolonged exposure of the 2CARD module.

An unexpected finding was that trapping the ATP state of RIG-I leads to a particularly increased interaction with the large ribosomal subunit via the expansion segment ES7L (*Figure 3D, F*). This expansion segment is present in metazoan ribosomes, however its length is substantially increased in human compared to drosophila ribosomes. The function of these expansion segments is not understood, but since helix E (ES7L-E) was recently found to interact with the selenoprotein synthesis factor SBP2, it is likely that the RNA in these elements is accessible to cytosolic proteins (*Kossinova et al., 2014*). The specific enrichment of the large ribosomal subunit under conditions where ribosomal subunits disengage argues for rather specific interactions of RIG-I E373Q with RNA present on the large but not the small subunit. The dominant binding of ribosomes by RIG-I E373Q can be explained by the high abundance of ribosomal RNA compared to other potential RIG-I ligands in the cytosol. We could directly visualize RIG-I E373Q on the ribosome at the solvent exposed root of ES7L-A (*Figure 3F, G*). This site contains a stretch of seven G:C/C:G base pairs, which approximately matches the footprint of dsRNA across the two SF2 RecA domains in the crystal structure of ADP·BeF$_x$-bound RIG-I (*Jiang et al., 2011*; *Kohlway et al., 2013*; *Kowalinski et al., 2011*; *Luo et al., 2011*) and also meets the requirements for activation of RIG-I's ATPase

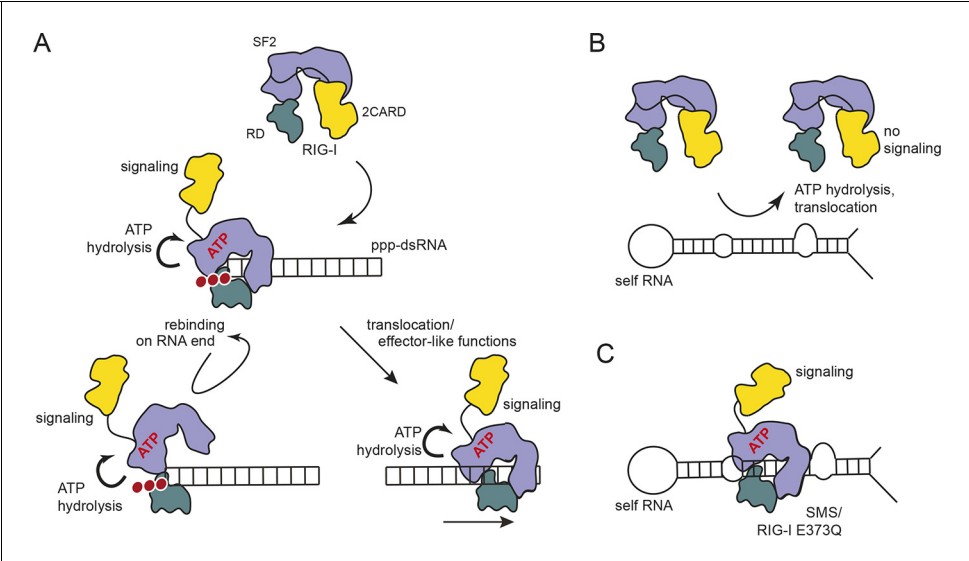

**Figure 6.** Proposed model for impact of ATP on RIG-I signaling on different RNAs. (**A**) RIG-I recognizes tri- or diphosphorylated double-stranded RNA and preferentially binds to the RNA end through its regulatory domain (RD, green). Binding of ATP-SF2 (purple) to the dsRNA releases the 2CARD module (yellow) and activates the downstream signaling process. ATP hydrolysis displaces the SF2 domain from dsRNA leading to either rebinding at the RNA end (tethered by RD) or to translocation along the RNA. (**B**) In healthy cells, sustained binding of RIG-I to self-RNA containing dsRNA stretches is prevented by ATP hydrolysis. The SF2 domain can be sufficiently displaced because the RD does not provide a high affinity tether. (**C**) Mutations that allow ATP promoted binding of dsRNA and displacement of the 2CARD module, but prevent ATP hydrolysis dependent dissociation of SF2 from dsRNA, such as those underlying atypical Singleton-Merten Syndrome, will result in an unintended signaling through self-RNA.

(*Anchisi et al., 2015*). Since 40% of the particles had this additional density, it is conceivable that additional binding sites could contribute to the interaction with RIG-I E737Q as well. However, the peripheral parts of the expansion segments are flexible and not visible in the 3D reconstructions, preventing us from observing RIG-I at other regions.

The RNA corresponding to the observed binding region of ES7L-A is also bound by RIG-I in vitro and can moderately stimulate RIG-I's ATPase (*Figure 4D*, *Figure 5A*). The much more efficient stimulation of RIG-I's ATPase by ppp-dsRNA is likely due to the high affinity towards RD, which could repeatedly "present" the RNA to SF2 (i.e. increasing the "local" concentration of RNA at SF2). Of note, while the addition of ATP to RIG-I reduces the interaction with the ES hairpin RNA, consistent with a role of the ATPase in preventing interaction with self-RNA, RIG-I E373Q binds with a moderately increased affinity to the ES-hairpin RNA in presence of ATP. Because of the large number of ribosomes in the cytosol it is therefore conceivable that RIG-I binds to double-stranded ribosomal RNA, including ES7L-A, under conditions where the ATPase is not able to efficiently displace the protein, such as those arising in patients with atypical SMS. In addition, the high local concentration of ribosomes in polysomes as well as a potential binding of RIG-I to other expansion segments could bring multiple RIG-I E373Q in contact, such that their exposed 2CARD module could interact for downstream signaling (*Peisley et al., 2014*; *Wu et al., 2014*). We do not, however, want to rule out contributions by other self-ligands as well. For instance, RIG-I can bind to endogenous mRNA (*Zhang et al., 2013*) or RNase-L cleavage products (*Malathi et al., 2007*), while MDA5 was shown to be activated by mRNA stem loop structures under conditions where reduction of A:T base-paired RNA is not prevented by ADAR1 (*Liddicoat et al., 2015*).

In any case, there are two levels of control to limit RLR mediated signaling to viral RNA. On one hand, RNA editing (*Liddicoat et al., 2015*) and methylation (*Schuberth-Wagner et al., 2015*) modifies particular types of self-RNA that would otherwise form reasonable ligands for RIG-I or MDA5. On the other hand, the intrinsic ATPase and translocase activity removes RLRs from short, but abundant endogenous dsRNA stretches, thereby reducing background signaling and increasing the sensitivity of the system.

## Materials and methods

### Cell lines, viruses and antibodies

Luciferase assays and RIG-I:RNA co-immunopurifications were carried out in HEK 293T cells (purchased from ATCC, CRL-11268) or HEK 293T RIG-I KO cells (*Zhu et al., 2014*). HEK 293T ISRE-FF/RFP reporter cells (stable expression of firefly luciferase and RFP under the control of an ISRE promoter, kindly provided by Luis Martinez-Sorbid, University of Rochester, Rochester, NY) were used for interferon stimulated luciferase reporter gene assays of recovered RNA. HEK cells were maintained in high glucose Dulbecco's Modified Eagle Medium supplemented with GlutaMAX, pyruvate and 10% FBS (all purchased from Gibco, UK). Human ribosomes were purified from HeLa S3 cells cultured in SMEM (Sigma, Germany) supplemented with 10% FBS, Penicillin (100 U/mL)/ Streptamycin (100 µg/mL) and 1x GlutaMAX (all purchased from Gibco, UK) using a spinner flask at 40 rpm. All cell lines were routinely checked for Mycoplasms by PCR and were, except for the HEK 293T ISRE-FF/RFP cell line, tested to be negative. Mycoplasm contaminations were suppressed using Plasmocin (InvivoGen, France) according to the manufacturer's protocol. Viruses used for infections were Sendai virus Cantell, Sendai virus defective interfering particles H4 (kindly provided by Dominique Garcin, Geneva, Switzerland), recombinant measles virus (MeV) with a sequence identical to the vaccine strain Schwarz (AF266291.1.)(*del Valle et al., 2007*; *Devaux et al., 2007*) and recombinant MeV-Cko-ATU-Cs. MeV-Cko-ATU-Cs expresses the C Schwarz protein from an additional transcription unit (ATU) located between the M and the P gene, while expression of C from the P gene is abrogated. Specifically, three stop codons were introduced into the P gene for the C ORF while leaving P and V protein expression intact. Cloning was done as described previously (*Pfaller and Conzelmann, 2008*; *Sparrer et al., 2012*). Additionally, an ATU was introduced between the P and M gene by duplicating the gene borders of the P gene. The ORF of the C (Schwarz) protein was cloned into that ATU and the virus rescued from cDNA using helper plasmids in 293-3-46 cells (*Radecke et al., 1995*) and propagated on Vero cells as described previously (*Parks et al., 1999*; *Pfaller et al., 2014*). Primary antibodies to human MDA5 (AT113) and RIG-I (Alme-1) were purchased from Enzo Life Science (Loerrach, Germany). Antibodies to FLAG (M2), HA (HA-7) and β-tubulin (TUB 2.1) were obtained from Sigma-Aldrich (Saint Luis, MO, USA). Secondary antibodies were supplied by GE Healthcare (Buckinghamshire, UK).

### Generation of RLR mutants

Sequences encoding full-length human RIG-I or MDA5 with N- or C-terminal FLAG/HA-tag were cloned into pcDNA5 FRT/TO (Invitrogen, Carlsbad, CA, USA). Mutants were generated by site-directed mutagenesis with PfuUltra polymerase (Agilent, Santa Clara, CA, USA).

### Immunoprecipitation of RLR-associated RNA from infected or non-infected cells

$6 \times 10^6$ HEK 293T or HEK 293T RIG-I KO cells were transfected with 10 µg pcDNA5 vector coding for different FLAG/HA tagged RLR proteins. Non-infected cells were harvested 24 h after transfection. Infections were carried out 6h after transfection with an MOI of 0.05 for measles virus or high MOI for Sendai virus and were allowed to proceed for 40 or 24 hr, respectively. Cells were harvested and incubated in Nonidet P-40 lysis buffer (50 mM HEPES, 150 mM KCl, 1 mM NaF, 0.5% NP-40, 0.5 mM DTT, protease inhibitor (Sigma, Saint Luis, MO, USA), pH 7.5) for 10 min on ice. Lysates were cleared by centrifugation and proteins were immunoprecipitated for 2.5 - –4 hr with anti-DDK magnetic beads (OriGene, Rockville, MD, USA) or anti-FLAG (M2) bound to magnetic protein G Dynabeads (Novex, Life Technologies, Carlsbad, CA, USA). Beads were washed five times with washing buffer (50 mM HEPES, 300 mM KCl, 0.05% NP-40, 0.5 mM DTT, protease inhibitor, pH 7.5) and incubated with proteinase K (Thermo Scientific, Vilnius, Lithuania) for 30 min at 50 °C. RNA was isolated by phenol/ chloroform/ isoamyl alcohol extraction using Phase Lock Gel Heavy tubes (5 PRIME, Germany). The quality of the isolated RNA was validated on an Agilent RNA 6000 Nano chip.

## Luciferase transfection assays

Immunoactivity experiments were carried out in 24-well plates seeded with $2.5 \times 10^5$ HEK 293T RIG-I KO or $2.5 \times 10^5$ HEK 293T ISRE-FF/RFP reporter cells per well using Lipofectamine 2000 (Invitrogen, Carlsbad, CA, USA) as transfection reagent according to the manufacturer's protocol. For downstream signaling assays HEK 293T RIG-I KO cells were co-transfected with 500 ng protein expression vector, 100 ng p125-luc, 10 ng pCMV-RL and 50 ng empty expression vector. For RIG-I E373Q/RIG-I Δ2CARD,E373Q competition assays HEK 293T RIG-I KO cells were co-transfected with 100 ng RIG-I E373Q expression vector, varying concentrations of the RIG-I Δ2CARD,E373Q expression vector, 100 ng p125-luc and 10 ng pCMV-RL. DNA concentrations were held constant by adding empty expression vector if necessary. For determination of the immunostimulatory potential of recovered RNA from co-immunoprecipitations, HEK 293T ISRE-FF/RFP cells were transfected with 250 ng RNA in Opti-MEM (Gibco, UK). For RNA stimulation of cells overexpressing RIG-I E373Q $2.5 \times 10^5$ HEK 293T RIG-I KO cells were transfected with 100 ng RIG-I E373Q expression vector, 100 ng p125-luc, 10 ng pCMV-RL and 1000 ng total RNA/ rRNA or ES hairpin RNA in Opti-MEM. All cells were harvested 24 h after transfection using 200 μL PLB (Promega, Madison, WI, USA) and subjected to immunoactivity experiments using the Dual-Glo luciferase assay system (Promega, Madison, WI, USA) as previously described (Runge et al., 2014). The luciferase activity was determined with a Berthold Luminometer in 96-well plates using 20 μL cell lysate.

## Protein expression and purification

RIG-I and RIG-I E373Q were expressed and purified from insect cells as described previously (Cui et al., 2008). Briefly, sequences encoding RIG-I were cloned into pFBDM vectors and transformed into *E. coli* DH10MultiBac cells. Bacmids were extracted for transfection into SF9 insect cells and propagated virus was used for protein expression in High Five insect cells. Seventy-two hours after infection cells were harvested and flash frozen in liquid nitrogen. RIG-I Δ2CARD was expressed in *E. coli* BL21 Rosetta (DE3), using pET expression vectors as described earlier (Cui et al., 2008). All recombinant proteins were purified using metal affinity (QIAGEN, Germany), heparin affinity and gel filtration chromatography (both GE Healthcare, Buckinghamshire, UK). Fractions containing RIG-I were concentrated to 6 mg/mL and flash-frozen in liquid nitrogen.

## Thermal unfolding assay

Thermal stability of RIG-I or RIG-I E373Q in presence or absence of ATP was analyzed by fluorescence thermal shift assays. Proteins (20 μM) were incubated in 25 mM HEPES pH 7, 150 mM NaCl, 10 mM MgCl$_2$, 5 mM TCEP, 5% glycerol and 5 mM ATP. After addition of SYPRO orange (Invitrogen, Carlsbad, CA, USA, final concentration: 2.5x) the fluorescence signal was detected using a gradient from 5 °C to 100 °C with 0.5 K/30 s and one scan each 0.5 K in a real-time thermal cycler (Biorad, Germany, CFX96 touch) using the FRET mode.

## Small-angle X-ray scattering

SAXS experiments were conducted at the PETRA3 P12 beamline of the European Molecular Biology Laboratory/ Deutsches Elektronen-Synchrotron, Hamburg, Germany. Samples were measured in absence or presence of 5 mM ATP in size exclusion buffer (25 mM HEPES pH 7, 150 mM NaCl, 5 mM MgCl$_2$, 5 mM β-Mercaptoethanol, 5% glycerol). RIG-I samples were measured at protein concentrations of 1.28, 2.65 and 8.35 mg/mL and RIG-I E373Q samples with concentrations of 0.87, 2.13 and 6.84 mg/mL. The respective scattering of the corresponding buffer was used for buffer subtraction. The samples did not show signs of radiation damage, which was assessed by automatic and manual comparison of consecutive exposure frames. The data was processed using PRIMUS from the ATSAS package (Konarev et al., 2006) and the radius of gyration was determined by Guinier plot [ln I(s) versus $s^2$] analysis obeying the Guinier approximation for globular proteins (s x R$_g$ < 1.3).

## Human 80S ribosome preparation

HeLa S3 cells were harvested (2 min, 650 x g), washed with PBS (Invitrogen, Carlsbad, CA, USA) and incubated with 1.5x vol Buffer 1 (10 mM HEPES/KOH, pH 7.2/4 °C, 10 mM KOAc, 1 mM Mg(OAc)$_2$ and 1 mM DTT) for 15 min on ice, followed by disruption with nitrogen pressure (300 psi, 30 min, 4 °C) in a cell disruption vessel (Parr Instrument, Moline, IL, USA). The cell lysate was cleared (10 min,

14,000 rpm, Eppendorf 5417R, 4 °C) and the resulting supernatant was loaded onto a sucrose cushion (Buffer 1 supplemented with 35% sucrose). Subsequent spinning (98 min, 75.000 rpm, TLA 120.2, 4 °C) was performed. After resuspension of the ribosomal pellet, a high-salt purification by centrifugation through a 500 mM sucrose cushion (50 mM Tris/HCl, pH 7.0/4 °C, 500 mM KOAc, 25 mM Mg(OAc)$_2$, 5 mM β-mercaptoethanol, 1 M sucrose, 1 µg/mL cycloheximide and 0.1% Nikkol) was conducted (45 min, 100,000 rpm, TLA120.2, 4 °C). The ribosomal pellet was resuspended in Ribosome Buffer (50 mM Tris/HCl, pH 7.0/4 °C, 100 mM KOAc, 6 mM Mg(OAc)$_2$, 1 mM DTT, 1/200 EDTA-free Complete protease inhibitor (Roche, Germany), 0.2 U/mL RNasin (Promega, Madison, WI, USA)), quickly centrifuged, frozen in liquid nitrogen and stored at -80 °C.

### Total RNA and ribosomal RNA isolation

For total RNA isolation 2.5 x 10$^5$ HEK 293T were seeded per well of 24 well plates. After 24 h cells were harvested in PBS, collected by centrifugation and lysed in Nonidet P-40 lysis buffer for 10 min on ice. Supernatant was cleared by centrifugation and DNA was digested with TURBO DNase (Ambion, Life Technologies, Carlsbad, CA, USA) for 3 min at 37 °C. Proteins were digested and RNA was extracted as described above. For ribosomal RNA isolation purified human ribosomes were proteinase K digested and RNA was extracted accordingly.

### Ribosomal binding studies

Human 80S ribosomes were incubated with or without 2.5x molar excess of RIG-I or RIG-I E373Q in binding buffer (50 mM HEPES/KOH, pH 7.5/ 4 °C, 100 mM KCl, 2.5 mM Mg(OAc)$_2$, 2 mM DTT, 1 mM ATP, 0.1% DDM, 10% Glycerol) for 15 min at room temperature and then for 15 min at 4 °C. The mixture was loaded onto a sucrose cushion (binding buffer with 750 mM sucrose) and spun (3 h, 40,000 rpm, SW55Ti, 4 °C). Supernatant and pellet fractions were separated and TCA precipitated. The resulting samples were analyzed by SDS-PAGE and visualized using SYPRO Orange Staining (Molecular Probes, Eugene, OR, USA).

### Cryo-grid preparation

5 OD/mL human 80S ribosomes were incubated with or without 2.5x molar excess of RIG-I E373Q. Each sample (50 mM HEPES / KOH, pH 7.5 / 4 °C, 100 mM KCl, 2.5 mM Mg(OAc)$_2$, 2 mM DTT, 1 mM ATP, 0.1% DDM, 5% glycerol) was applied to 2 nm pre-coated Quantifoil R3/3 holey carbon supported grids and vitrified using a Vitrobot Mark IV (FEI Company , Germany).

### Cryo-electron microscopy and single particle reconstruction

Data were collected on a 120 keV TECNAI SPIRIT cryo-electron microscope with a pixel size of 2.85 Å/pixel at a defocus range between 1.4 µm and 4.6 µm (with RIG-I E373Q ligand) or between 1.8 µm and 5.3 µm (without ligand) under low dose conditions. Particles were detected with SIGNATURE (*Chen and Grigorieff, 2007*). Initial alignment resulted in 61,067 particles (with ligand) and 29,959 particles (without ligand). Subsequent data processing and single particle analysis was performed using the SPIDER software package (*Frank et al., 1996*). Non-ribosomal particles (19,080 particles, 31% (with ligand) and 10,663 particles, 35% (without ligand)) were removed from each data set by unsupervised 3D sorting (*Loerke et al., 2010*). The remaining particles were further sorted, resulting in a volume with additional density (with ligand: 23,715 particles, 39% ). The identical sorting scheme was applied to the control 80S ribosome without ligand, resulting in final 11,727 particles (39% ). The final 80S structures with and without ligand were refined to an overall resolution (FCS$_{0.5}$) of 17.7 Å and 21.9 Å, respectively. For comparison of the two final volumes, a statistical difference map between the two reconstructions was calculated.

### Figure preparations and model docking

We used the crystal structure of the human RIG-I protein (PDB code 3TMI) (*Jiang et al., 2011*) and the human ribosome (PDB 4V6X) (*Anger, et al., 2013*) for rigid-body fitting into the additional density. Figures depicting atomic models with and without density were prepared using UCSF Chimera (*Pettersen et al., 2004*).

## Differential radial capillary action of ligand assay

ATP binding was determined by DRaCALA using [$\alpha$-$^{32}$P]ATP (Hartmann Analytik, Germany). 12 µM RIG-I or RIG-I E373Q were incubated in 50 mM HEPES, pH 7.5, 150 mM KCl, 5 mM MgCl2, 2.5 mM TCEP, 0.1 mg/mL BSA supplemented with 2.5 nM [$\alpha$-$^{32}$P]ATP for 10 min at room temperature in presence or absence of 100 nM RNA. 2.5 µL of reaction mixture was spotted on nitrocellulose membranes (0.22 µM pores, GE Healthcare, Buckinghamshire, UK), air-dried and [$\alpha$-$^{32}$P]ATP was detected using a phosphor-imaging system (GE Healthcare, Germany).

## Electrophoretic mobility shift assay

Proteins at different concentrations were pre-incubated with ATP, ADP or ADP·BeF$_3$ (all 3 mM end concentration, ADP·BeF$_3$ was generated using ADP, NaF and BeCl$_2$ in a 1:1:5 molar ratio) and added to 0.5 µM ES hairpin RNA or 0.2 µM 24mer RNA in EMSA buffer (50 mM Tris pH 7.5, 50 mM KCl, 5 mM MgCl$_2$, 5 mM TCEP, 7.5 µM ZnCl$_2$, 3 mM ATP, 5% glycerol). Reactions were incubated for 20 min at 37 °C. Samples were separated on TB agarose gels (89 mM Tris, 89 mM boric acid, 0.8% agarose) and stained with Gel-Red (Biotium, Hayward, CA, USA). Unbound RNA bands were quantified with ImageJ.

## Fluorescence anisotropy assays

Different RIG-I or RIG-I E373Q protein concentrations were titrated into EMSA buffer without ATP and glycerol. Reactions were started by addition of 5 mM ATP and 20 nM Cy3- or Cy5-labeled RNA and fluorescence anisotropy was measured with a TECAN M1000 plate reader after incubation at room temperature for 20 min.

## ATPase hydrolysis assays

ATPase hydrolysis activity was determined using [$\gamma$-$^{32}$P]ATP (Hartmann Analytik, Germany). Proteins at different concentrations were pre-incubated with 100 nM RNA or purified ribosomes for 10 min at room temperature in EMSA buffer without ATP. The reaction was initiated by addition of 1.5 mM unlabeled and 10 nM [$\gamma$-$^{32}$P]ATP and incubated for 20 min at 37 °C. Free phosphate was separated from ATP by thin layer chromatography in TLC running buffer (1 M formic acid, 0.5 M LiCl) on polyethyleneimine cellulose TLC plates (Sigma-Aldrich, Germany). [$\gamma$-$^{32}$P]P$_i$ and [$\gamma$-$^{32}$P]ATP were detected using a phosphor-imaging system (GE Healthcare, Germany) and quantified using ImageJ.

# Acknowledgements

We thank Simon Runge for help with RNA co-purification during initial stages of the project and Filiz Civril for the RIG-I insect cell expression vector. We thank Luis Martinez-Sorbido (University of Rochester, Rochester, NY) for the HEK 293T ISRE-FF/RFP cell line and Dominique Garcin (University of Geneva, Switzerland) for providing SeV defective interfering particles H4. We thank Stefan Krebs and Andrea Klanner for support in RNA analysis and Andrea Gilmozzi for help with ribosome purification and binding assays. Furthermore, we thank the staff of the EMBL/DESY PETRA3 P12 beamline for support in SAXS measurements, Gregor Witte for help with SAXS data analysis, Tobias Deimling for help with EMSAs and ATPase assays and Robert Byrne for critically reading the manuscript.

# Additional information

### Competing interests

GH, VH co-founder and shareholder of the Rigontec GmbH. The other authors declare that no competing interests exist.

### Funding

| Funder | Grant reference number | Author |
| --- | --- | --- |
| National Institutes of Health | T32 training grant 5T32AI007647-13 | Jenish R Patel |
| German Excellence Initiative | CIPSM | Roland Beckmann Karl-Peter Hopfner |

| | | |
|---|---|---|
| Graduate School of Quantitative Biosciences Munich | | Roland Beckmann Karl-Peter Hopfner |
| European Research Council | Advanced Grant CRYOTRANSLATION | Roland Beckmann |
| Deutsche Forschungsgemeinschaft | FOR 1805 | Roland Beckmann |
| Deutsche Forschungsgemeinschaft | SFB646 | Roland Beckmann Karl-Peter Hopfner |
| Deutsche Forschungsgemeinschaft | GRK1721 | Karl-Peter Hopfner Roland Beckmann |
| Bavarian network for Molecular Biosystems | | Karl-Peter Hopfner |

The funders had no role in study design, data collection and interpretation, or the decision to submit the work for publication.

### Author contributions

CL, Planned and performed luciferase assays and co-IP experiments, purified proteins, performed in vitro assays and wrote the paper; SM, Purified human ribosomes, carried out ribosome binding studies and analyzed the cryo-EM structure; KMJS, Produced measles virus and planned the luciferase assays and co-IPs; CCdOM, Carried out co-IP experiments and thermal shift assays; MM, Purified proteins; JRP, Provided Sendai virus Cantell; MG, GH, Provided synthetic 24mer and 12mer ppp-RNAs; AGS, K-KC, RB, Supervised the study and critically read the manuscript; VH, Provided the 293T RIG-I KO cell line and critically read the manuscript; K-PH, Designed the research, supervised experiments and wrote the paper

### Author ORCIDs

Karl-Peter Hopfner, http://orcid.org/0000-0002-4528-8357

## Additional files

### Major datasets

The following previously published datasets were used:

| Author(s) | Year | Dataset title | Dataset URL | Database, license, and accessibility information |
|---|---|---|---|---|
| Jiang F, Ramanathan A, Miller MT, Tang GQ, Gale M, Patel SS, Marcotrigiano J | 2011 | Structural basis of RNA recognition and activation by innate immune receptor RIG-I | http://www.rcsb.org/pdb/explore/explore.do?structureId=3TMI | Publicly available at the RCSB Protein Data Bank (Accession no: 3TMI). |
| Anger AM, Armache JP, Beringhausen O, Habeck M, Subklewe M, Wilson DN, Beckmann R | 2013 | Structures of the human and Drosophila 80S ribosome | http://www.rcsb.org/pdb/explore/explore.do?structureId=4V6X | Publicly available at the RCSB Protein Data Bank (Accession no: 4V6X). |

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
