## [Decision Letter]

Thank you for submitting your work entitled "RIG-I's ATPase prevents unintentional recognition of self-RNA" for peer review at *eLife*. Your submission has been favorably evaluated by John Kuriyan (Senior editor), a Reviewing editor, and three reviewers.

The reviewers have discussed the reviews with one another and the Reviewing editor has drafted this decision to help you prepare a revised submission.

Summary:

RIG-I is a cytoplasmic viral sensor that induces antiviral IFN responses by recognizing non-self signatures in viral RNAs, such as 5' triphosphate and dsRNA structure. Recently, the correlation between autoimmunity and mutations in RIG-I family molecules has been reported. However, the mechanism of how mutant RIG-I leads to autoimmune symptoms is not well characterized. Here Lassig et al. investigated RIG-I mutations found in patients with Singleton-Merten Syndrome (SMS) and confirmed that the SMS mutant RIG-I (C268F and E373A) constitutively activate signaling. Moreover, they found that another constitutively active E373Q mutant is ATPase-deficient and exhibits prolonged binding with endogenous RNAs such as 28S rRNA, which does not activate wild-type RIG-I. These data suggest that abnormal immune response to endogenous RNA is a possible cause of SMS in patients with RIG-I C268F and E373A mutations. The authors also propose that ATP hydrolysis prevents unintentional recognition of self-RNA. The work is highly significant and the data are generally of high quality with some exceptions noted below.

This manuscript contains important observations on RIG-I SMS mutants. However, to support their conclusion that constitutive activity of SMS mutants and E373Q mutant is due to hyper-responsiveness to endogenous RNAs, the following major concerns need to be addressed.

Essential revisions:

1) The role of ATP binding versus ATP hydrolysis:

ATPase deficient RIG-I proteins cannot be activated by viral infection and act as dominant inhibitors. Consequently, in paragraph four, subheading “RIG-I ATP hydrolysis defective mutant E373Q shows increased interaction with ribosomal RNA”, the interpretation of Figure 2 needs to be expanded. It is not clear how this data is supportive of the major conclusion from the paper. It seems that RIG-I should be observed bound to the 80S RNA in the absence of ATP, but it is not observed (i.e., ATP hydrolysis is expected to dislodge RIG-I from this RNA). How does this data fit with the idea that ATP hydrolysis helps to remove RIG-I from the 80S RNA because it appears that RIG-I does not bind to this RNA, in the presence or absence of ATP?

In this regard, one important control lacking in the current manuscript is the characterization of the wt RIG-I in the presence of a non-hydrolysable analog of ATP such as ATP-gamma-S or AMPPNP. It would be meaningful to show, e.g., by EMSA, that wt RIG-I binds 28S rRNA in the presence of one of these analogs. This wound make it clear that ATP binding is required for binding RNA, yet ATP hydrolysis is needed to unbind cellular RNA.

2) Activation by rRNA:

It is known that binding to RNAs does not necessarily activate RIG-I. Thus, how can authors prove that 28S rRNA activates RIG-I E373Q for IFN signaling in the situation where no ATPase activity can be detectable in this mutant? For instance, can RIG-I E373Q stably expressing cells in RIG-I deficient cells respond to exogenous stimulation with rRNAs?

In this regard, it will be helpful to add some more explanation about the nature of the 28S rRNA. How does it dominate the RNA binding site of E373Q mutant? Does the relative abundance/concentration amongst similarly structured RNA explain it? Is it the particular secondary structure of this RNA that makes it outcompete amongst other RNA molecules?

3) The possibility that constitutive signaling exhibited by the SMS mutant and the E373Q mutant is due to a structural alteration that opens and exposes the CARD domain:

When the RIG-I E373Q mutant responds to endogenous RNAs, how does it undergo the conformational change to elicit CARDs without ATPase activity? The authors should exclude the possibility that the mutant protein assumes an "active" conformation simply due to the amino acid substitutions without the need to bind to endogenous ligand, i.e., due to the constitutive exposure of CARD. It might be possible to resolve this important issue by using a construct of E373Q with the CARDs deleted; this construct should inhibit E373Q full-length mediated signaling in a dominant negative fashion because this mutant should bind to endogenous ligand but fail to trigger signaling due to the absence of CARDs.

Major editorial details to address:

1) The title of this manuscript is misleading since some known ATPase deficient mutants such as K270A lose their potential to activate IFN signaling in response to any ligand, and ATPase activity is required for wild-type RIG-I to transduce the conformational changes needed for signaling in response to viral dsRNA in a physiological condition. Thus, these well-established observations are contradictory to their conclusion as appears in the title of the manuscript.

2) In Figure 3, it would be appropriate to show quantification of the gel bands. It's also not clear what the 24 dsRNA and 12 dsRNA represent in the context of manuscript. They are definitely counterparts of 5ppp-dsRNA, which mimic viral RNA. Then do the 24bp and 12bp dsRNA represent cellular RNA or the less stimulatory part of viral RNA?

3) The quality of T347A,E373Q mutant protein was not provided. Assays performed in Figure 3 should be tested for the T347A,E373Q mutant protein as well.

---

## [Author Response]

Essential revisions:

1) The role of ATP binding versus ATP hydrolysis:ATPase deficient RIG-I proteins cannot be activated by viral infection and act as dominant inhibitors. Consequently, in paragraph four, subheading “RIG-I ATP hydrolysis defective mutant E373Q shows increased interaction with ribosomal RNA”, the interpretation of Figure 2 needs to be expanded. It is not clear how this data is supportive of the major conclusion from the paper. It seems that RIG-I should be observed bound to the 80S RNA in the absence of ATP, but it is not observed (i.e., ATP hydrolysis is expected to dislodge RIG-I from this RNA). How does this data fit with the idea that ATP hydrolysis helps to remove RIG-I from the 80S RNA because it appears that RIG-I does not bind to this RNA, in the presence or absence of ATP?In this regard, one important control lacking in the current manuscript is the characterization of the wt RIG-I in the presence of a non-hydrolysable analog of ATP such as ATP-gamma-S or AMPPNP. It would be meaningful to show, e.g., by EMSA, that wt RIG-I binds 28S rRNA in the presence of one of these analogs. This wound make it clear that ATP binding is required for binding RNA, yet ATP hydrolysis is needed to unbind cellular RNA.

As suggested, we performed EMSAs with RIG‐I and the ribosomal expansion segment derived hairpin RNA in presence of the non‐hydrolysable ATP analog ADP∙BeF_3_ and added the data to the manuscript (Figure 4). We also added EMSAs with RIG‐I and the 24mer dsRNA in presence of ADP∙BeF_3_ (Figure 4). In addition we repeated our ribosome-binding assay in order to better visualize binding of RIG‐I in absence of ATP and added the ADP∙BeF_3_ control (Figure 3). We find that RIG‐I in complex with ADP∙BeF_3_ behaves like RIG‐I E373Q in complex with ATP, adding this important control raised by the referees. We also carefully repeated the ribosome-binding assay and could confirm binding of RIG‐I as well as RIG‐I E373Q in the absence of ATP. We thank the referees for raising these questions. The new data strengthen our model and we added these data to the Results (paragraph three of the subsection “RIG-I ATP hydrolysis defective mutant E373Q shows increased interaction with ribosomal RNA” and paragraphs two and three of “Specificity of RIG-I towards double-stranded RNA is increased in presence of ATP”).

2) Activation by rRNA:It is known that binding to RNAs does not necessarily activate RIG-I. Thus, how can authors prove that 28S rRNA activates RIG-I E373Q for IFN signaling in the situation where no ATPase activity can be detectable in this mutant? For instance, can RIG-I E373Q stably expressing cells in RIG-I deficient cells respond to exogenous stimulation with rRNAs?In this regard, it will be helpful to add some more explanation about the nature of the 28S rRNA. How does it dominate the RNA binding site of E373Q mutant? Does the relative abundance/concentration amongst similarly structured RNA explain it? Is it the particular secondary structure of this RNA that makes it outcompete amongst other RNA molecules?

We thank the referees for this suggestion. We performed co‐transfection experiments of RIG‐I E373Q with purified total RNA, ribosomal RNA and the expansion segment derived hairpin RNA and in all cases find an increase in signaling. These data are added to Figure 2.

Even a weak activation of RIG‐I E373Q by 28S rRNA can lead to a considerable signal due to the huge amount of ribosomal RNA compared to other RNAs in the cytosol. Nevertheless we do not want to exclude that there are other double‐stranded RNAs that can activate RIG‐I E373 and stated this in the Discussion (paragraphs four and five).

3) The possibility that constitutive signaling exhibited by the SMS mutant and the E373Q mutant is due to a structural alteration that opens and exposes the CARD domain:When the RIG-I E373Q mutant responds to endogenous RNAs, how does it undergo the conformational change to elicit CARDs without ATPase activity? The authors should exclude the possibility that the mutant protein assumes an "active" conformation simply due to the amino acid substitutions without the need to bind to endogenous ligand, i.e., due to the constitutive exposure of CARD. It might be possible to resolve this important issue by using a construct of E373Q with the CARDs deleted; this construct should inhibit E373Q full-length mediated signaling in a dominant negative fashion because this mutant should bind to endogenous ligand but fail to trigger signaling due to the absence of CARDs.

To rule out a “constitutive” active conformation due to exposed CARDs we performed 3 experiments. We did small angle X‐ray scattering on RIG‐I and RIG‐I E373Q and demonstrate that both proteins have the same solution structure (Figure 1—figure supplement 2). We also performed thermal unfolding assays and find that the E373Q mutation does not destabilize the protein (Figure 1—figure supplement 2). Finally, we performed the suggested competition assay, demonstrating that RIG‐I Δ2CARD E373Q has a dominant negative effect on signaling by RIG‐I E373Q (Figure 1 and Figure 1—figure supplement 2). All three experiments confirm that RIG‐I E373Q is not constitutively active due to exposure of CARDs because of ATP binding or unfolding, and are included in paragraph three of the subsection “Prevention of ATP hydrolysis in RIG-I leads to a constitutive activation of the interferon-β promoter by recognition of self-RNA”.

Major editorial details to address:1) The title of this manuscript is misleading since some known ATPase deficient mutants such as K270A lose their potential to activate IFN signaling in response to any ligand, and ATPase activity is required for wild-type RIG-I to transduce the conformational changes needed for signaling in response to viral dsRNA in a physiological condition. Thus, these well-established observations are contradictory to their conclusion as appears in the title of the manuscript.

We thank the editor for this valuable advice and we changed the title to: “ATP hydrolysis by the viral RNA sensor RIG‐I prevents unintentional recognition of self‐RNA”.

2) In Figure 3, it would be appropriate to show quantification of the gel bands. It's also not clear what the 24 dsRNA and 12 dsRNA represent in the context of manuscript. They are definitely counterparts of 5ppp-dsRNA, which mimic viral RNA. Then do the 24bp and 12bp dsRNA represent cellular RNA or the less stimulatory part of viral RNA?

We substantially revised our EMSAs, repeated them at least in triplicates and quantified the bands (Figure 4 and Figure 4—figure supplement 1).

We used different RNA types (triphosphorylated double‐stranded RNA, blunt‐ended double‐stranded RNA and hairpin RNA with a non‐pairing end) in order to dissect differences in binding to the RNA end by RD vs. binding of the SF2 domain to the stem. We revised the text to clarify the use of different RNAs (first paragraph of “Specificity of RIG-I towards double-stranded RNA is increased in presence of ATP”).

3) The quality of T347A,E373Q mutant protein was not provided. Assays performed in Figure 3 should be tested for the T347A,E373Q mutant protein as well.

As suggested we purified RIG‐I T347A,E373Q and performed EMSAs with the ES hairpin RNA in presence and absence of ATP as well as ATPase assays (paragraphs three and four, subection “Specificity of RIG-I towards double-stranded RNA is increased in presence of ATP”; Figure 4, Figure 4—figure supplement 1 and Figure 5—figure supplement 1). The data nicely demonstrate that T347A reduces ES hairpin RNA binding of the E373Q mutant.